# Parametric Aerodynamic Study of Galloping Piezoelectric Energy Harvester with Arcuate Protruding and Depressed Features

**DOI:** 10.3390/s25061657

**Published:** 2025-03-07

**Authors:** Xiaokang Yang, Bingke Xu, Zhendong Shang, Chunyang Liu, Haichao Cai, Xiangyi Hu

**Affiliations:** 1School of Mechatronics Engineering, Henan University of Science and Technology, Luoyang 471002, China; xbk1026@163.com (B.X.); jdszd@haust.edu.cn (Z.S.); chunyangliu@haust.edu.cn (C.L.); chc1226@haust.edu.cn (H.C.); hxy230626@haust.edu.cn (X.H.); 2Longmen Laboratory, Luoyang 471002, China

**Keywords:** energy harvesting, piezoelectricity, galloping, aerodynamic parameter

## Abstract

This study explores the potential effect of a cross-sectional shape with an arcuate protruding and depressed features on the performance. The geometric configurations include two feature types (protruding and depressed), each with six distinct perimeter arrangements and three depths per arrangement, yielding thirty-six different cross-sectional shapes for systematic evaluation. The aerodynamic characteristics and electrical performance are numerically analyzed, using a computational fluid dynamics model and a distributed parameter electromechanical coupling model, respectively. A smooth protruding feature on the front, top, or bottom side suppresses the electrical output; however, when located on the rear side, it significantly increases the slope of the power versus wind speed curve. Depressed features on the rear, top, or bottom side only reduce the critical wind speed and the power enhancement positively correlates with the feature depth. Compared to a square, a harvester with depressed feature on both top and bottom sides exhibits a significant jump in power at the critical wind speed, greatly improving the power. These findings provide important design guidelines for structural optimization of galloping piezoelectric energy harvesters, enabling them to match the wind energy distribution characteristics of specific regions with optimal performance.

## 1. Introduction

Driven by the demand for environmental monitoring applications in large-span and off-grid areas, micro energy harvesters based on wind-induced vibration, such as vortex-induced vibration, galloping, and wake galloping, have attracted much attention [1,2,3]. These devices can provide a continuous energy supply for wireless sensor nodes or networks, thereby reducing maintenance costs [4]. Galloping is a typical aerodynamic instability phenomenon with large amplitude, low frequency, and self-excitation oscillations, which is beneficial for designing a galloping-based piezoelectric energy harvester (GPEH), as the electrical output of a piezoelectric transducer is positively correlated to the oscillation amplitude [5,6]. In recent years, GPEHs have emerged as a dynamic research direction in the field of micro power technology.

In the field of civil engineering, significant effort has been dedicated over the past few decades to both theoretical and experimental studies on the galloping behavior of various bluff bodies with different cross-sectional shapes [7]. One reason supporting such research is the widespread construction of buildings with innovative cross-sectional shapes, whose critical conditions for aerodynamic instability need to be properly designed [8]. In contrast to civil engineering researchers who strive to avoid aerodynamic instability, the primary goal of the structural design of the GPEH is to achieve optimal performance by adapting to the wind energy distribution characteristics of a specific area.

Initially, the research on the influence of cross-sectional shape on performance mainly focused on bluff bodies with simple shapes, such as square [9], rectangular [10], triangular [11,12], elliptical [13,14], D-shaped [9], V-shaped [15,16], W-shaped [17], Y-shaped [18], fork-shaped [19], etc. An experimental study by Yang et al. [9] confirmed the superiority of the square cross-sectional shape compared to the rectangular, triangular, and D-shaped in terms of output power, with a maximum power of 8.4 mW being achieved. Roohi et al. [13] proposed a GPEH with an elliptical bluff body and examined the impact of the aspect ratio and mass of the bluff body on the harvested power. Their results demonstrated a notable decrease in extracted power density for aspect ratio of 1 and 2, compared to higher aspect ratios. Siriyothai et al. [16] focused on the performance enhancement of a GPEH with different V-shaped groove depths on a square bluff body. The results showed that the system with a groove depth to the side ratio of 0.25 generated the highest power of 15.24 mW, which is 1.34 times higher than the power generated by the square bluff body at 9 m/s.

GPEHs with complex cross-sectional shapes have also been widely explored in recent years, such as sinusoidal wavy protuberances on squares [20], circulars with different rod-shaped attachments [21,22], and rectangular, triangular, and elliptical metasurfaces’ protrusions [23]. Ewere et al. [20] proposed and experimentally investigated a GPEH with a sinusoidal wavy leading edge on a square cross-sectional bluff body, inspired by the tubercles of a humpback whale flipper. The results showed that these protuberances could be used as a passive control to tune the critical galloping velocity. Hu et al. [21] studied the performance of wind energy harvesters with rod-shaped attachments. The experimental results showed that the energy capture performance of the energy harvesters with triangular rods was better than that of the other two groups due to the larger transverse force coefficient. Wang et al. [22] proposed a new piezoelectric energy harvester with a Y-shaped attachment (GPEH-Y). The vibration form of the GPEH-Y transformed from vortex-induced vibration to galloping as the wind speed increased. Compared with the piezoelectric energy harvester based on vortex-induced vibration with a smooth cylinder, GPEH-Y had a higher energy harvesting efficiency. Xing et al. [23] proposed three types of protruded bluff bodies with rectangular, triangular, and elliptical metasurfaces, and four kinds of surface treatments were deployed to vary their protruded shape. Wind tunnel experiments revealed that adding the protrusions can obviously change the mode of oscillations, and only the backward protrusions can enhance the galloping response. The maximum output power with a backward protruded length of 15 mm in the experiments was measured to be 0.757 mW, which occurs at 5.1 m/s, and the energy harvester outperformed its counterpart carrying a simple square prism by 157.48%. A common characteristic of these studies is the use of irregular feature variations in the cross-sectional shape, which results in non-universal conclusions, making it difficult to design structures that meet specific performance requirements.

Once the aerodynamic characteristics of bluff bodies with different cross-sectional shapes are obtained through experiments or simulations, an accurate mathematical model is essential to analyze the effect of the aerodynamic parameters on the performance [1]. A single-degree-of-freedom lumped parameter model was developed in early studies. It can only predict the first-order vibration modal response of the system, and the system parameters need to be identified through experimental methods. The distributed parameter model based on Rayleigh–Ritz discretization or Euler–Bernoulli beam theory can provide more accurate analysis. Abdelkefi et al. [24] established a coupled nonlinear electroaeroelastic distributed parameter model based on Euler–Bernoulli beam theory, which was validated by the previous experimental results of Sirohi and Mahadik [25]. The effects of the load resistance and wind speed on the performance of the GPEH have been studied, and the results showed that the onset speed is strongly affected by the electrical load resistance. Zhao et al. [26] developed a comparative study on the performance of the modeling methods, including the single-degree-of-freedom lumped parameter model and the single-mode and multi-mode Euler–Bernoulli distributed parameter models. The influence of the load resistance, wind exposure area, mass of the bluff body, and length of the piezoelectric sheets on the cut-in wind speed, as well as the output power level, were fully investigated. It was demonstrated that all the considered models could accurately predict the performance of the harvester.

In this study, the protruding and depressed features of different depths of features around the square bluff body are considered separately, with the aim of investigating the effects of the cross-sectional shape with continuous feature variations on the performance of the GPEH. In Section 2, a simplified distributed parameter model, which was proposed in our previous work, is validated by the experimental results of a prototype with a square cross-sectional shape. Computational fluid dynamics is employed to perform the time-varying simulation of aerodynamic characteristics for thirty-six bluff bodies with different cross-sectional shapes and the effects of aerodynamic parameters on performance are analyzed using the model in Section 3. A summary and conclusions are presented in Section 4.

## 2. Modeling and Validation

The common structural design of the GPEH is where one end of the piezoelectric composite beam is connected to the bluff body, and the other end is fixed, as shown in Figure 1. According to the quasi-steady theory, the lift and drag coefficients of galloping, *C_L_* and *C_D_*, in the course of oscillation, are the same at each angle of attack, *α,* as the values measured at the corresponding steady angle of attack [27]. The bluff body with special and non-axisymmetric cross-sectional shape will experience galloping instability, if the Den Hartog criterion [28] is satisfied, that is, the slope of the *C_L_* is more negative than the *C_D_* when *α =* 0°. The galloping transverse force *F_y_* and the criterion are expressed as follows:(1)$$F_{y}=-\frac{1}{2}\rho u^{2}{LDC}_{Fy}=\frac{1}{2}\rho u^{2}LD\left(C_{L}+C_{D}tan\alpha \right)sec\alpha ,$$(2)$${\left.\frac{\partial C_{Fy}}{\partial \alpha }\right|}_{\alpha =0^{{}^{\circ}}}=-{\left.\left(\frac{\partial C_{L}}{\partial \alpha }+C_{D}\right)\right|}_{\alpha =0^{{}^{\circ}}}>0,$$
where *C_Fy_* represents the galloping aerodynamic coefficient; *L* and *D* represent the height and length of the bluff body, respectively; and *u* and *ρ* represent the wind velocity and airflow density, respectively.

The aerodynamic coefficient *C_Fy_* is primarily determined by the cross-sectional shape, turbulence intensity, and Reynolds number. When the boundary layer is turbulent rather than laminar, it is better able to remain attached to the bluff body surface and flow separation is delayed which reduces the pressure drag significantly. It is well known that for a bluff body, compressive stress is the dominant component of the aerodynamic drag force, rather than the wall shear stress. For instance, Laneville [29] conducted experiments on a square cross-sectional bluff body, and the results indicated that the effect of increasing turbulence intensity is to progressively reduce the maximum lift, while the initial trend at small angles of attack remains unchanged. This reduction is accompanied by a shift in the angle of attack corresponding to the maximum lift toward smaller angles. A similar shift toward smaller angles is observed at the minimum drag. At small angles of attack, an increase in turbulence intensity leads to a decrease in drag, whereas beyond the angle of minimum drag, no significant intensity effect is observed. Ultimately, the initial trend of the *C_Fy_* versus the angle of attack curve remains relatively unchanged at small angles, but both the peak value of the *C_Fy_* and its corresponding angle of attack decrease with higher turbulence intensity. Consequently, the galloping displacement amplitude of the bluff body shows moderate reduction. In natural environments, GPEHs operating near the ground are typically influenced by turbulent flow rather than laminar flow. As a result, the cross-sectional shape of the bluff body becomes the most crucial factor affecting the aerodynamic characteristics. The protruding and depressed features of different depths around the square bluff body are considered separately. The bluff body length, *D*, is 60 mm, and three different feature depths, *d*, are considered, which are 30, 20, and 10 mm, as shown in Figure 1. As the ratio of *d/D* decreases, the cross-sectional shape tends to resemble a square. This approach embodies the logic of the way in which a complex cross-sectional feature evolves toward the simplest and most readily obtainable shape according to a certain pattern. This makes it easier to capture the effect of the cross-sectional shape with the continuous feature variations on aerodynamic characteristics. The final shape is named “${ROUGH}_{F/R/FR/T/B/TB/FRTB}^{i-P/D}$”, with the superscript and subscript denoting the feature depth, and the direction of the protruding (*P*) or depressed (*D*) features, respectively. For example, ${ROUGH}_{FRTB}^{30-P}$ means a square cross-sectional shape with the arcuate protruding features of 30 mm depth on the front (*F*), rear (*R*), top (*T*), and bottom (*B*) sides.

A distributed-parameter electromechanical coupling model of a piezoelectric energy harvester based on the interaction between vortex-induced vibration and galloping has been proposed and verified [30]. The distributed-parameter model of the GPEH can be obtained under non-short-circuit conditions and by neglecting the aerodynamic force of vortex-induced vibration [31]. The first-order model can then be expressed as follows:(3)$$\ddot{Y}\left(t\right)+2\omega \xi \dot{Y}\left(t\right)+\omega^{2}Y\left(t\right)+\Theta V\left(t\right)=\frac{1}{2}\rho u^{2}LDC_{Fy}$$(4)$$\frac{V\left(t\right)}{R}+C_{p}\dot{V}\left(t\right)=\Theta \dot{Y}\left(t\right)$$
where ω, ξ, and Θ represent the undamped natural frequency, mechanical damping ratio, and piezoelectric coupling term, respectively; V represents the voltage across the load resistor R; and $C_{p}$ represents the total capacitance.

The aerodynamic coefficient *C_Fy_* is a function of the angle of attack *α*, and therefore of $tan\left(\alpha \right)=\dot{Y}/u$. A polynomial of $\dot{Y}/u$ can be used to approximate this function, as follows:(5)$$C_{Fy}=\sum_{i=0}^{N}A_{i}{\left(\frac{\dot{Y}}{u}\right)}^{i}$$
where the symbol *A_i_* represents the coefficient in the polynomial. Note that if the cross-sectional shape is symmetric in the downstream direction, the polynomial will be an odd function.

The prototype was fabricated and tested in a small wind tunnel, as shown in Figure 2a. The dimensions of the bluff body and the piezoelectric composite beam are 4 × 4 × 10 mm and 14 × 4 × 0.52 mm, respectively. The theoretical critical wind speed of galloping is expressed as $u_{g}=\frac{4m\omega \xi }{\rho LDA_{1}}$, where *m* represents the equivalent mass. Under the premise of a minuscule volume, a relatively smaller mass and resonant frequency are advantageous for achieving a lower critical wind speed. Low-density balsa wood (Wanxu Balsawood Trading Co. Ltd., Guangzhou, China) and the flexible polyvinylidene fluoride film (TE Connectivity, Schaffhausen, Switzerland) have been selected for the fabrication of the bluff body and the beam, respectively. The mass of the bluff body is 0.0291 g. The beam is assembled using a 20 μm thick ultraviolet-curing process to attach 200 μm thick PVDF films to 300 μm thick polyethylene terephthalate film (Toray Industries, INC., Tokyo, Japan). According to the model, the numerical first natural frequency, capacitance, and optical load are 97.29 Hz, 132.8 pF, and 12.32 MΩ, respectively. The experimental values are 98.8 Hz, 130.5 pF, and 13 MΩ, as determined using the amplitude–frequency characteristic, impedance analyzer, and load characteristic, respectively, as shown in Figure 2b,c. The numerical results for these parameters agree well with the experimental ones and the relative errors are approximately 1.6%, 1.7%, and 5.5%, respectively. The numerical equivalent mass and piezoelectric coupling term are 0.03082 g and 1.935 × 10^−6^ N/V, respectively. The total damping ratio, ξ, was measured using the logarithmic decrement method [25], with values ranging from 1.60% to 1.87%. The total damping ratio comprises contributions from structural damping, aerodynamic damping, and electrical damping. Due to the relatively low piezoelectric strain constant of PVDF, the resulting electrical damping ratio contribution is relatively minor. According to the experimental studies by Huang et al. [32] on a square cross-section structure with modified corners, the influence of corner modifications on aerodynamic damping can be neglected, although aerodynamic damping is dependent on wind speed. A variable damping ratio was applied to the calculation process, and a linear fit was used to express the relationship between damping and wind speed, as follows: ξ=0.00123u+0.0044.

The time marching using ODE45 is used to obtain the numerical solutions of the displacement or electrical responses and the flow chart of the simulation process is shown in Figure 3a. The convergence condition necessitates that the amplitude error of the aerodynamic coefficient or displacement response of the bluff body between two adjacent time steps must be less than 10^−5^, or until the calculation time reaches its maximum value. An updated initial amplitude should be considered, which means that the initial amplitude of each velocity calculation point is taken to be equal to the stable amplitude of the previous one, because the hysteresis characteristics of the galloping would not be predicted by considering a fixed initial amplitude. The experimental results and numerical results with fixed (1.74%) and variable damping ratios are shown in Figure 3b. The critical wind speed of the experiment is 9.2–9.4 m/s. Once the wind speed exceeds the critical wind speed, the load obtains a larger electrical response. For example, when the wind speeds are 10, 12, and 13.2 m/s, the experimental output powers are 23.1, 26.3, and 27.4 μW, respectively. In the working wind speed range, the experiment power versus wind speed curve is divided into linear and nonlinear regions, and the wind speed ranges are 9.4–11.2 m/s and 11.2–13.2 m/s, respectively. The nonlinear response region may be caused by material and geometric nonlinearity at the relatively high wind speeds. According to previous studies, the geometric nonlinearity of flexible cantilever beam systems under large displacements should be considered for its impact on bluff body motion [33]. This nonlinear strain–deflection relationship will lead to an amplitude reduction trend. Additionally, the piezoelectric material nonlinearity can induce a softening behavior in the system [34]. Therefore, higher-order elastic and electroelastic tensor components in the piezoelectric constitutive equations should be considered during the modeling process. The current model exhibits discrepancies in predicting electrical responses under high wind speeds compared to experimental results, primarily due to the exclusion of nonlinear factors in the modeling framework.

With the variable damping ratio, the model accurately predicts the critical wind speed and the electrical response level of the linear region. Since the model does not consider nonlinear factors, the numerical output powers under high wind speed are higher than the experimental values. For example, when the wind speed is 13.2 m/s, the numerical and experimental output powers are 33.7 (fixed damping ratio), 31.1 (variable damping ratio), and 27.4 μW, respectively, with relative errors of 23.0% and 12.4%. In addition, the prototype has a slight electrical output below the critical wind speed, which may be caused by the non-strict symmetry structure of the prototype. In 2013, Yang et al. [9] proposed a prototype of the GPEH with a square cross-sectional shape, and a single-degree-of-freedom lumped parameter model was used for the prediction of the electrical response after the identification of the prototype’s parameters. The equivalent mass, natural frequency, damping ratio, piezoelectric coupling term, capacitance, and optimized load are 30.168 g, 6.84 Hz (short circuit) and 6.8 Hz (open circuit), 1.48% (0.5% was used), 0.000373 N/V, 180 nF, and 105 kΩ, respectively. Figure 3c shows a comparison of the predicted results from the current model with the numerical and experimental and simulation data in Yang’s study, and the results of the two models are almost the same in terms of critical wind speed and output voltage.

To obtain the aerodynamic force on the bluff body, the external flow field is simulated by solving the continuity and the Navier–Stokes equations of incompressible fluid, with the assumption that the external flow field is 2*D* and unsteady [12,23]. As shown in Figure 4a, the computational domain is rectangular with a size of 70*D* × 20*D*, where *D* is the characteristic dimension of the cross-section of the bluff body. The distances from the center point of the bluff body to the inlet and top wall are 20*D* and 20*D*; these distances ensure adequate fluid development and a smaller blocking rate. The settings of the boundary conditions include the velocity inlet, pressure outlet, and symmetric top and bottom walls. A circular mesh refinement area around the bluff body is set, with a diameter of 5*D*. The large eddy simulation (LES) model can adequately capture the turbulence details, but this comes at a high computational expense and strong mesh quality dependence, especially the strict grid resolution requirements at high Reynolds numbers [35]. The characteristic length of the blunt body is 0.06 m, and its Reynolds number is approximately 12,322. Considering computational efficiency and stability, the standard k−ε model was selected. According to boundary layer theory, turbulence near the wall is not fully developed. Wall functions are used to handle the near-wall flow issues to ensure the model effectiveness. In this case, the first grid layer needs to be positioned within the fully developed turbulent flow region, namely the log-law region. The height of the first grid layer is determined by the y+ value which is close to 30 [36]. The time step in the domain-independence study is finally set to 0.0005 s, meeting the requirement of the maximum Courant number to be less than unity.

The reference values are used in the computation of the derived physical quantities and non-dimensional coefficients for postprocessing, such as the lift and drag force coefficients. The accuracy of the calculation results is related to the reference value setting. The reference values, which are defined in ANSYS Fluent^®^ 2023 R1(ANSYS, Inc., Southpointe, PA, USA), mainly include the area, depth, length, and velocity. The three dimensional dimensions of the bluff body are uniquely determined by the area, depth, and length. The depth is the thickness of the bluff body, which corresponds to the *L* in this paper and its value also needs to be set in 2D simulations. The length corresponding to the *D* in this paper refers to the projection dimension of the bluff body in the downstream directions. In order to facilitate the understanding of the area definition, a temporarily introduced height variable refers to the projection dimension of the bluff body in the cross-flow direction, which corresponds to the *B* in this paper. Finally, the area is the projection of the bluff body in the cross-flow direction, which is equal to the product of the projection dimensions of the height and depth. It should also be noted that the area definition does not apply to the slender structures, such as airfoils, where drag force is primarily caused by the wall shear stress rather than the compressive stress. In this case, the area is defined as the projection of the blunt body in the downstream direction. Three different grid densities are considered to ensure the grid independence. The comparisons of results of the aerodynamic lift and drag coefficients, *C_L_* and *C_D_*, in the time domain are given in Figure 4b,c, when the angle of attack, wind speed, Reynolds number, and turbulence intensity are 0°, 15.3 m/s, 20,000, and 12.5%, respectively. The root mean square (RMS) values of *C_L_* and *C_D_* are given in Table 1, and are consistent with the experimental results in Laneville’s study [29], and the medium grid number has been selected.

The numerical results for *C_L_*, *C_D_*, and *C_Fy_* of the thirty-six different cross-sectional shapes are shown in Figure 5 and Figure 6, in which the protruding and depressed features are considered. Due to the asymmetric structure in the cross-flow direction, both the positive and negative angles of attack are considered, with a range of −30–30°. The aerodynamic characteristic analysis includes the protruding features on the front and rear sides and the depressed features on the top and bottom sides of the cross-sectional shape.

## 3. Results and Discussion

### 3.1. Aerodynamic Characteristics

#### 3.1.1. Protruding and Depressed Features on the Front and Rear Sides

As shown in Figure 5a, unlike the sharp edges of the square shape, the smooth protruding feature on the front side only results in a positive slope for the lift coefficient *C_L_* curve at an angle of attack *α* of 0°, and the *C_L_* increases with the *α*. Additionally, protruding features on both the front and rear sides with a larger feature depth have profiles that are closer to a circular, resulting in an average lift coefficient that is almost zero. As shown in Figure 5c, the slope of the *C_Fy_* versus *α* curve is non-positive at *α* = 0°, and no galloping behavior occurs for ${ROUGH}_{F}^{30-P}$, ${ROUGH}_{F}^{20-P}$, ${ROUGH}_{FR}^{30-P}$, and ${ROUGH}_{FR}^{20-P}$. As shown in Figure 5b, the *C_D_* of the protruding features on the front side only and both the front and rear sides decreases with increasing feature depth, and their values are lower than those of the square at all considered *α*. A notable difference is observed with the protruding features on the rear side only, where the *C_D_* is higher than that of the square at *α* below 10°, but lower than that of the square at *α* greater than 10°. The rear side protruding feature has a significant effect on the *C_L_* when the *α* exceeds 10°. At this point, although its drag coefficient *C_D_* is lower than that of the square, it still causes the peak value of the aerodynamic coefficient *C_Fy_* versus the *α* curve and the *α* corresponding to *C_Fy_* = 0 is clearly larger than the square. According to the previous research conducted by Xing et al. [23] and Hu et al. [21], the larger peak value reflects a greater galloping aerodynamic force and displacement response, which is beneficial to the performance improvement of the GPEH.

The depressed feature, as shown in Figure 5d–f, does not have a significant effect on *C_L_* and *C_D_*, and these minor effects are ultimately not reflected in *C_Fy_*. When *α* is greater than 10°, the depressed feature on the rear side suppresses the variation in *C_D_*. When *α* is below 10°, the depressed feature on the front side results in a higher *C_D_* compared to the square. However, at smaller angles of attack, the contribution of *C_D_* to *C_Fy_* is relatively small, so the values of *C_Fy_* at all the considered *α* for the depressed feature are nearly identical to those of the square.

#### 3.1.2. Protruding and Depressed Features on the Top and Bottom Sides

Due to the fact that the shapes with protruding and depressed features on the top or bottom side are only asymmetric in the cross-flow direction, the coefficient curves of *C_L_*, *C_D_*, and *C_Fy_* versus the α of these shapes are not symmetric with the origin point of coordinate axis in the range −30° < α < 30°. Therefore, for those cases, the effects of the protruding and depressed features on *C_L_*, *C_D_*, and *C_Fy_* can only be discussed in the range 0° < α < 30°. For the protruding feature, as shown in Figure 6a–c, the *C_L_* and *C_D_* values are higher and lower than those of the square, respectively, at most α and are mainly caused by the smooth top or bottom protruding feature. This results in both the slope of *C_Fy_* versus *α* at *α* = 0° and the peak value of the *C_Fy_* being lower than those of the square shape. The subtle variation in the *C_Fy_* leads to a relatively weak galloping effect on the bluff body, characterized by a higher critical wind speed and a smaller displacement amplitude. An important difference is shown by the depressed feature, as shown in Figure 6d–f. The values of *C_L_* and *C_D_* with the top-side only depressed feature are almost the same as those of the square in 0° < α < 30°. This is because when α is greater than 0°, the top depressed feature is located on the rear side of the shape and has almost no effect on *C_L_* and *C_D_*. The bottom depressed features have limited effects on *C_L_* and *C_D_* at a smaller α (such as 0–6°), but their effects in a larger *α* (such as 6–30°) are obvious. According to the results, it can be seen that the degree of influence is positively correlated with the depth of the depressed feature. For example, as shown in Figure 6f, the peak values of *C_Fy_* and the *α* corresponding to *C_Fy_* = 0 of the depressed feature on both the top and bottom sides with the feature depths 30, 20, and 10 and the square shape are 0.88 (10°), 0.55 (10°), 0.37 (10°), 0.20 (6°), and 18–20°, 16–18°, 14–16°, 10–12°, respectively.

In conclusion, an asymmetric shape with the depressed feature on the top or bottom side only is necessary, because they have little effect on the lift and drag coefficient when those features are located at the rear of the shape due to the angle of attack. The results of the depressed feature on the rear side only also support this view. The degree of influence is positively correlated with the depth of the depressed feature; therefore, the case of ${ROUGH}_{TB}^{30-CV}$ has a larger galloping displacement response, and the GPEH with this cross-sectional shape has a much higher electrical output.

### 3.2. Electrical Performance of GPEH

The polynomial coefficient *A_i_* of all the considered cross-sectional shapes is obtained using the curve fitting method, as shown in Table A1, Table A2, Table A3, Table A4, Table A5, Table A6, Table A7, Table A8 and Table A9 in Appendix A, and the R-squared value is configured to be no less than 0.997 to ensure that the fitted line explains most of the variability of the response data around its mean. The parameters used in the model are the same as those of the prototype in Section 2, except for the *A_i_*. The wind speed step and the maximum calculation time are configured to 0.2 m/s and 100 s, respectively. The convergence condition is that the amplitude error of the displacement response between two adjacent time steps must be less than 10^−5^, or until the calculation time reaches its maximum value. In previous studies, critical wind speed and maximum power have commonly been used as key performance indicators, representing the operational wind speed range and the ultimate performance level, respectively. From an application perspective, a fundamental principle should be followed, that is, the harvester should be designed to optimally match the spatial and temporal distribution characteristics of wind energy. This means that once the wind speed exceeds the critical wind speed, the harvester should quickly transition to its optimal operating state. Conversely, solely pursuing lower or ultra-low critical wind speeds may not necessarily be the most suitable approach. Based on this, the slope of the power versus wind speed curve should also be introduced as one of the performance evaluation parameters. A quadratic function is employed to fit the power versus wind speed relationship beyond the critical wind speed. The slope of the power versus wind speed curve is derived by differentiating the function with respect to wind speed.

As shown in Figure 7a–c, when the GPEH with a smooth front side protrusion feature experiences weak galloping or no galloping at all, its power output is strongly suppressed. In contrast, the rear side protrusion feature positively enhances the output power, with the enhancement level being positively correlated with the depth of the protrusion, as shown in Table 2. For instance, the ${ROUGH}_{R}^{30-P}$ achieves a maximum output power 5.86 times that of ${ROUGH}_{square}$ at 16 m/s. This enhancement effect of the posterior protrusion on output power is similar to the findings of Xing et al. [23] Their experimental study demonstrated that a prototype with a 15 mm elliptical rear protrusion achieved a peak power of 0.757 mW at 5.1 m/s, surpassing the square by 157.48%. As shown in Figure 7d–f, the slope of the power versus wind speed curve of the GPEH with the depressed feature is higher than that of square and the maximum slope coefficient observed in ${ROUGH}_{FR}^{20-D}$, where the equation is *k* = 1.84*u*−8.85. Furthermore, for the depressed feature on the front or rear side only, the critical wind speed decreases as the feature depth increases. In particular, introducing a rear side depressed feature is an effective method for achieving a GPEH with a lower critical wind speed compared to square. For example, the critical wind speeds of ${ROUGH}_{R}^{30-D}$, ${ROUGH}_{R}^{20-D}$, and ${ROUGH}_{R}^{10-D}$ are 19.15%, 19.15%, and 14.89% lower than that of ${ROUGH}_{square}$, respectively. Finally, the optimal performance is achieved in ${ROUGH}_{F}^{10-D}$ and ${ROUGH}_{FR}^{20-D}$. ${ROUGH}_{F}^{10-D}$ has a critical wind speed 6.38% lower than that of the square, while ${ROUGH}_{FR}^{20-D}$ has the same critical wind speed as the square but exhibits the largest slope coefficient across all cases and then at a wind speed of 16 m/s; the power is 96.17 and 96.25 µW, respectively.

The effects of the protruding and depressed features on the top and bottom sides on the performance are shown in Figure 8. The protruding features on the top or bottom side only have a significant effect on the critical wind speed and the slope of the power versus wind speed curve which decreases and increases with the increase in the feature depth, and the performance with those features is inferior to that of the square, as shown in Figure 8a. For the protruding features on both the top and the bottom sides, no case experiences galloping oscillation, as shown in Figure 8b. This shows that the configuration of the smooth protruding feature on the top or bottom side makes no positive contribution to the improvement of the performance of the GPEH. As shown in Figure 8c, the slope of the power versus wind speed curve for all considered features is higher than that of the square, and the depressed feature on the top or bottom side only contributes to a reduction in the critical wind speed, as shown in Table 3. As shown in Figure 8d, although the critical wind speed of a GPEH with both the top and bottom sides’ depressed features is higher than that of the square, a significant sudden leap of the power versus wind speed curve is observed at the critical wind speed. This is because the depressed features cause a larger peak value and an angle of attack corresponding to *C_Fy_* = 0, which, in turn, produces a higher galloping displacement and electrical response amplitude. For example, the powers of ${ROUGH}_{TB}^{30-D}$, ${ROUGH}_{TB}^{20-D}$, and ${ROUGH}_{TB}^{10-D}$ are 288.3, 192.9, and 28.5 μW, respectively, which is about 5.59, 4.60, and 9.38 times that of the square at the same wind speed.

Based on the aforementioned conclusions and the requirements of specific application scenarios, the structural design process of the GPEH can adhere to the following guidelines. In wind sparse regions, a low critical wind speed is essential, while in wind abundant regions, a steep power versus wind speed curve is crucial. Practical constraint by operational space limitation results in a finite system volume. According to the theoretical critical wind speed of galloping, a smaller equivalent mass, damping ratio, and resonant frequency, along with a larger coefficient *A_1_*, all contribute to a lower critical wind speed. Consequently, the GPEH’s bluff body and cantilever beam are typically fabricated using lightweight materials and flexible piezoelectric materials, respectively. Among all the configurations considered in this study, those with a depressed feature on the rear, top, or bottom side only exhibit a higher coefficient *A_1_* than the square, which in turn leads to a lower critical wind speed. Moreover, a GPEH with both the top and bottom sides’ depressed features has a noticeable leap at the critical wind speed and a steeper slope in the power versus wind speed curve, leading to superior power performance compared to the square.

## 4. Conclusions

In summary, this work explored the potential effect of the cross-sectional shape with continuous feature variations on the performance of a galloping piezoelectric energy harvester (GPEH). It proved that the protruding and depressed features of different feature depths around the square bluff body can significantly change the aerodynamic characteristics, galloping behavior, and electrical performance. Thirty-six considered cross-sectional shapes were simulated using computational fluid dynamics and an experimentally verified model was used to evaluate the performance level. From an application perspective, in addition to the critical wind speed and maximum power, the slope of the power–wind speed curve was introduced as one of the performance evaluation parameters, used to assess the rate at which the harvester reaches its optimal operating state. There are some remarkable findings in this work, as follows:i.A GPEH with a smooth arcuate protrusion feature on the front, top, or bottom side experienced weak galloping or no galloping at all, and its power output was strongly suppressed.ii.Introducing a depressed feature on the rear, top, or bottom side only was an effective method for achieving a GPEH with a lower critical wind speed compared to the square and the power enhancement level positively correlated with the feature depth.iii.Although the critical wind speed of a GPEH with both the top and bottom sides depressed feature was higher than that of the square, a noticeable leap in the power versus wind speed curve was observed at the critical wind speed, which significantly enhanced the GPEH’s performance.

In future work, alternative shapes such as triangles, trapezoids, and others could be employed to replace the arcuate design and Boolean operations can be applied to understand the graphic construction of the cross-sectional shape. Additionally, a 3D computational fluid dynamics model can be considered to capture the intricate details of the flow field.

## Figures and Tables

**Figure 1 sensors-25-01657-f001:**
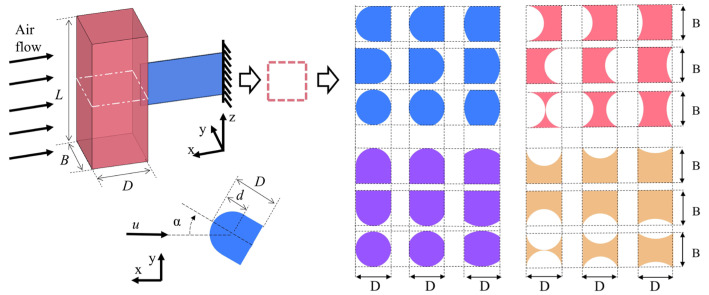
Schematic of the GPEH and the configuration of its cross-sectional shape.

**Figure 2 sensors-25-01657-f002:**
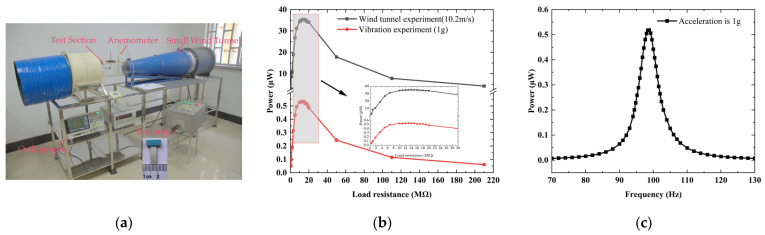
(**a**) Experimental setup; (**b**) power versus load; (**c**) power versus frequency.

**Figure 3 sensors-25-01657-f003:**
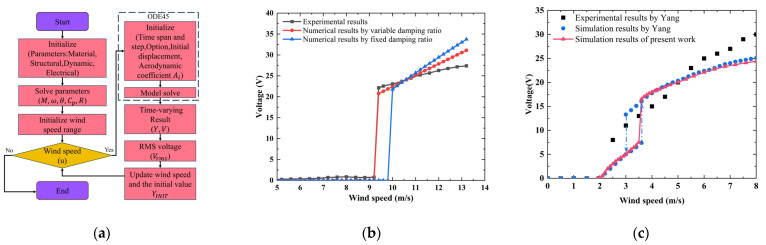
(**a**) Flow chart of the simulation process; (**b**) experimental and numerical results for the prototype; (**c**) comparison of results from the proposed model and the previous study by Yang [9].

**Figure 4 sensors-25-01657-f004:**
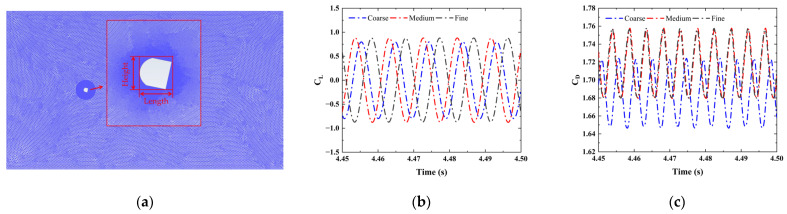
(**a**) Computational mesh; comparison of the results in time domain: (**b**) *C_L_*, (**c**) *C_D_*.

**Figure 5 sensors-25-01657-f005:**
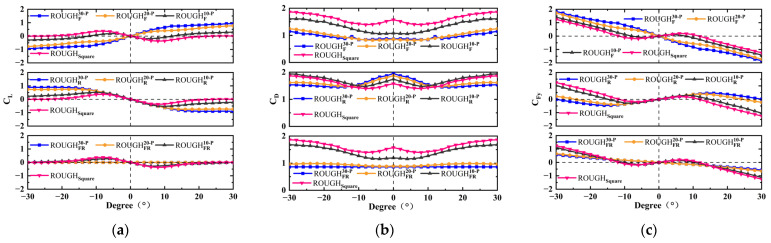
Numerical coefficients versus the angle of attack for protruding and depressed features on the front and/or rear sides: (**a**) *C_L_*, (**b**) *C_D_*, and (**c**) *C_Fy_* for the protruding feature; (**d**) *C_L_*, (**e**) *C_D_*, and (**f**) *C_Fy_* for the depressed feature.

**Figure 6 sensors-25-01657-f006:**
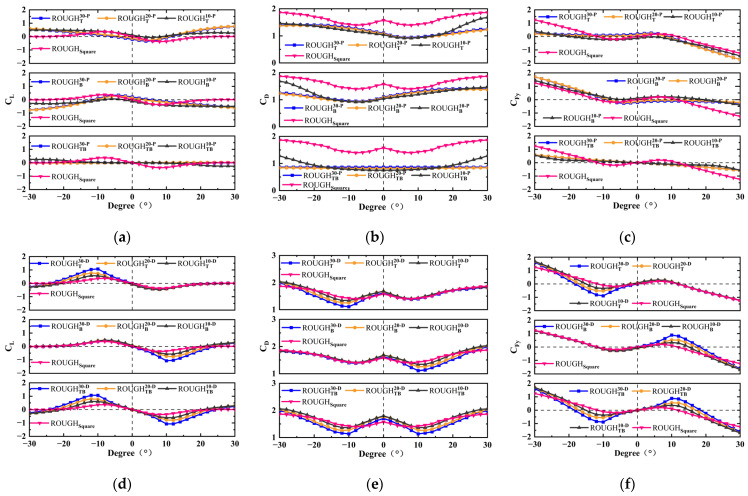
Numerical coefficients versus the angle of attack for protruding and depressed features on the top and/or bottom sides: (**a**) *C_L_*, (**b**) *C_D_*, and (**c**) *C_Fy_* for the protruding feature; (**d**) *C_L_*, (**e**) *C_D_*, and (**f**) *C_Fy_* for the depressed feature.

**Figure 7 sensors-25-01657-f007:**
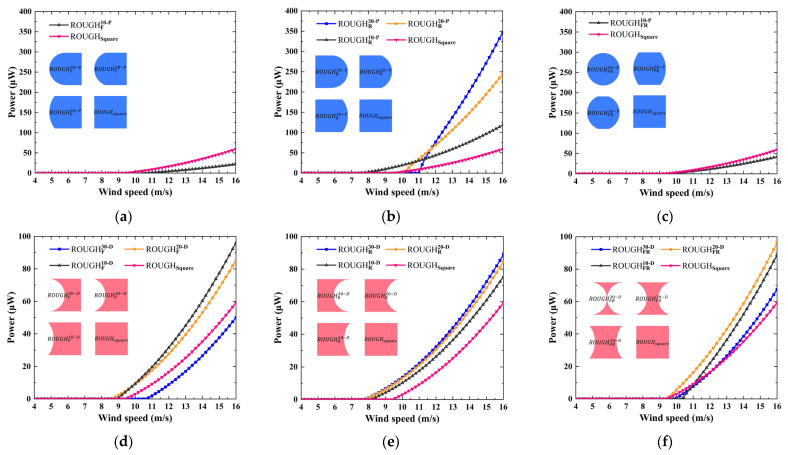
Numerical power versus wind speed: protruding features on (**a**) the front side only, (**b**) the rear side only, and (**c**) both the front and rear sides; depressed features on (**d**) the front side only, (**e**) the rear side only, and (**f**) both the front and rear sides.

**Figure 8 sensors-25-01657-f008:**
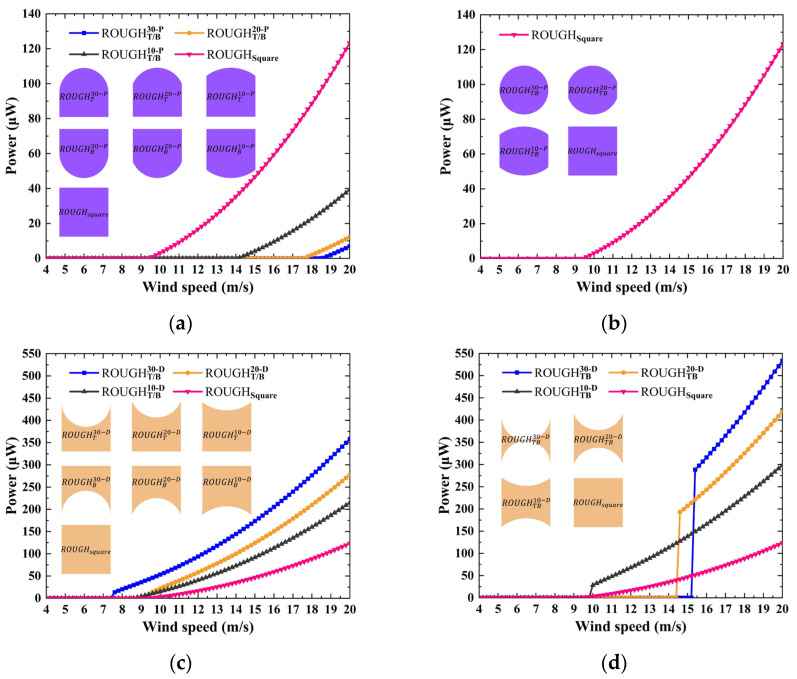
Numerical power versus wind speed: protruding features on (**a**) the top or bottom side only, and (**b**) both the top and bottom sides; depressed features on (**c**) the top or bottom side only, and (**d**) both the top and bottom sides.

**Table 1 sensors-25-01657-t001:** Comparison of the results for *C_L_* and *C_D_*.

Case	Grid Number	*C_L(RMS)_*	*C_D(RMS)_*
Coarse	$3.72\times 10^{4}$	0.566	1.683
Medium	$1.03\times 10^{5}$	0.622 [9.89%]	1.716 [1.90%]
Fine	$2.34\times 10^{5}$	0.624 [0.32%]	1.716 [0.00%]

**Table 2 sensors-25-01657-t002:** Performance for the rear side protrusion feature and the square.

Scenario	Critical Wind Speed (m/s)	Maximum Power at 16 m/s (μW)	Slope of the Power Versus Wind Speed Curve
${ROUGH}_{R}^{30-P}$	11.0	347.37	*k* = 6.50*u* − 22.53
${ROUGH}_{R}^{20-P}$	10.0	245.08	*k* = 4.75*u* − 21.73
${ROUGH}_{R}^{10-P}$	7.4	118.37	*k* = 2.76*u* − 11.44
${ROUGH}_{square}$	9.4	59.29	*k* = 1.31*u* − 7.78

**Table 3 sensors-25-01657-t003:** Performance for the top and bottom sides’ depressed features and the square.

Case	Critical Wind Speed (m/s)	Maximum Power at 16 m/s (μW)	Slope of the Power Versus Wind Speed Curve
${ROUGH}_{T/B}^{30-D}$	7.4	205.37	*k = 2.47u* − *6.59*
${ROUGH}_{T/B}^{20-D}$	9.2	150.62	*k = 1.97u* − *4.07*
${ROUGH}_{T/B}^{10-D}$	8.6	112.79	*k = 1.79u* − *6.90*
${ROUGH}_{TB}^{30-D}$	15.2	315.89	*k = 3.78u* − *13.52*
${ROUGH}_{TB}^{20-D}$	14.4	243.23	*k = 2.97u* − *9.50*
${ROUGH}_{TB}^{10-D}$	9.8	165.62	*k = 2.06u* − *4.23*
${ROUGH}_{square}$	9.4	59.29	*k* =1.31*u* − 7.78

## Data Availability

Data are contained within the article.

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
