# Peer review of "Parametric Aerodynamic Study of Galloping Piezoelectric Energy Harvester with Arcuate Protruding and Depressed Features"

_sensors, 2025, doi:10.3390/s25061657_

Round 1

Reviewer 1 Report

Comments and Suggestions for Authors

line 147 the influence of turbulence on the results is not discussed in detail, but there are no detailed data on the possible influence of turbulence on the lift (CL) and drag (CD) coefficients

line 179 It is worth noting how the dimensions, type of materials used, and configuration of the installation will affect the results.

line 398 there is a lack of conclusion about the practical application of this work, about possible limitations in scaling

Reviewer 2 Report

Comments and Suggestions for Authors

The manuscript presents a parametric aerodynamic study on galloping piezoelectric energy harvesters (GPEH) with varied cross-sectional bluff body shapes. The work employs computational fluid dynamics (CFD) and distributed parameter models to analyze aerodynamic characteristics and electrical performance. While the study is comprehensive and addresses an important topic in energy harvesting, several revisions are required to improve clarity, methodology description, and result interpretation. Specific comments are outlined below:

  1. The title emphasizes "parametric aerodynamic study" but does not explicitly mention key design factors such as "arcuate protruding/depressed features" or "critical wind speed optimization," which are central to the work.
  2. The abstract lacks a clear statement of the research gap and practical significance. The phrase "continuous feature variations" is vague and requires elaboration.
  3. The CFD setup (e.g., mesh independence study, turbulence model selection) and boundary conditions are insufficiently detailed. The rationale for using the standard k-ε model instead of more accurate transient models (e.g., LES) is unclear.
  4. The damping ratio is described as amplitude-dependent and fitted linearly to wind speed, but the physical basis for this correlation is not explained.
  5. Figure 3(b) shows discrepancies between numerical and experimental power outputs at high wind speeds (e.g., 23% error at 13.2 m/s). The authors attribute this to neglected nonlinearities but do not quantify or discuss these effects.The "sudden leap" in power for depressed top/bottom features (Figure 8(d)) lacks a mechanistic explanation.
  6. Figures 5–8 lack axis labels with units, making interpretation difficult.Table 2 and 3 report slope equations (e.g., k = 6.50u - 22.53) without defining k or explaining their derivation.
  7. In Table 2, ROUGH<sub>R</sub><sup>30-P</sup> achieves 347.37 μW at 16 m/s, while ROUGH<sub>square</sub> yields only 59.29 μW. This sixfold improvement is remarkable but lacks validation against prior studies (e.g., Xing et al. [23]).
  8. The conclusion does not address limitations (e.g., 2D CFD assumptions, lack of 3D flow effects) or future work.
  9. References [21] and [23] are cited in the text but omitted from the reference list.
  10. Minor errors exist, e.g., "ROUGH<sub>FR</sub><sup>20−D</sup>" in Section 4 (page 10) lacks consistent subscript/superscript formatting.
Comments on the Quality of English Language

The English could be improved to more clearly express the research.
